# Hydration and Mechanical Properties of High-Volume Fly Ash Concrete with Nano-Silica and Silica Fume

**DOI:** 10.3390/ma15196599

**Published:** 2022-09-23

**Authors:** Byung-Jun Kim, Geon-Wook Lee, Young-Cheol Choi

**Affiliations:** Department of Civil and Environmental Engineering, Gachon University, Seongnam 13120, Gyeonggi-do, Korea

**Keywords:** cement composite, filler effect, microstructure, nano-silica, silica fume

## Abstract

This study investigated the effects of nano-silica (NS) and silica fume (SF) on the hydration reaction of high-volume fly ash cement (HVFC) composites. In order to solve the dispersibility problem caused by the agglomeration of NS powder, NS and NSF solutions were prepared. NS content and SF content were used as main variables, and an HVFC paste was prepared in which 50% of the cement volume was replaced by fly ash (FA). The initial heat of hydration was measured using isothermal calorimetry to analyze the effects of NS and SF on the initial hydration properties of the HVFC. In addition, the compressive strength was analyzed by age. The refinement of the pore structure by the nanomaterial was analyzed using mercury intrusion porosimetry (MIP). The results show that the addition of NS and SF shortened the setting time and induction period by accelerating the initial hydration reaction of HVFC composites and improved the compressive strength during the initial stage of hydration. In addition, the micropore structure was improved by the pozzolanic reaction of NS and SF, thereby increasing the compressive strength during the middle stage of hydration.

## 1. Introduction

With ongoing industrial development and coal consumption, the importance of research on carbon neutrality in the construction industry has increased. Accordingly, studies have been conducted on a variety of construction materials, and the use of supplementary cementing materials (SCMs), such as fly ash (FA), ground granulated blast-furnace slag (GGBS), and silica fume (SF), has attracted attention [1,2,3]. SCMs are known to contribute to the engineering properties of cement composites through the reaction of such pozzolans with cement [4,5,6,7].

Of these SCMs, FA has been widely used as a cement admixture because of its low heat of hydration, shrinkage-reducing effect, and high durability. Roshani et al. [8] predicted the effect of fly ash on the mechanical properties of concrete using artificial neural networks (ANN). Studies have been actively researching high-volume fly ash cement (HVFC) containing large amounts of FA. HVFC reduces greenhouse gas emissions by decreasing cement consumption and has major benefits in terms of the recycling of industrial byproducts. However, it has low initial compressive strength because FA remains inactive during the initial hydration process [4,9,10,11,12]. To address this problem, many processes have been studied, including the grinding of FA [13], chemical activation [14], mechanochemical treatment [15], and hydrothermal treatment [16,17]. Yang et al. [18] reported that adding SF to Na_2_SO_4_-activated high-volume fly ash (HVFA) concrete leads to higher resistance and less strength loss. Bondar et al. [19] reported that the initial strength of HVFA paste was improved by the use of cement kiln dust (CKD) and gypsum as activators.

In recent years, there has been growing interest in research to improve the performance of cementitious materials, including the use of nanomaterials to improve the initial strength of HVFA concrete. In general, nanomaterials accelerate the hydration of cement and improve its microstructure, thereby enhancing its mechanical performance and durability [20,21,22]. The last ten years has seen rapid development of nanomaterials in various fields. For example, nanomaterials that can transform concrete have appeared, such as nano-SiO_2_ (NS) and nano-TiO_2_ [23,24,25,26]. Onaizi et al. [27] used nano-sized waste glass powder to improve the early strength of HVFA concrete. They reported that the increase in compressive strength decreased when nano glass powder was added more than 10%.

Nano-silica (NS) is a material with very small particle sizes and the characteristics of a pozzolanic material. The dissolved silica component reacts with the calcium hydroxide generated from the cement hydration reaction to form additional calcium silicate hydrate (C-S-H) and contribute to strength development [28,29,30,31,32,33,34,35,36,37,38,39,40,41,42,43,44,45,46,47,48,49,50,51,52,53,54,55,56,57]. Therefore, many studies on concrete have been conducted using NS. Karakouzian et al. [58] studied the effect of carbon nanotube and NS on the mechanical properties of concrete. They reported that NS can improve not only the early strength of concrete, but also the long-term strength by the pozzolan effect. Kong et al. [35] investigated the effect of NS on the microstructure and mechanical properties of cementitious materials. They found that resistance to calcium- leaching and mechanical performance were improved regardless of the type or amount of NS. Li [28] reported that NS can improve the low initial strength of HVFC by accelerating the cement hydration reaction. The application of NS to HVFC, however, may have a negative impact on long-term strength. Kawashima et al. [38] reported that an NS content of 5% or more decreased the long-term strength of HVFC as the pozzolanic reaction products of NS interfered with the reaction of FA particles.

As NS may form weak zones in the matrix because of its agglomeration characteristics, NS dispersibility is an important matter of concern [59]. An excessive amount of NS may also affect strength negatively by decreasing the fluidity of the concrete. Ghafari et al. [50] investigated the effect of dry NS powder on the fluidity, strength, and transport performance of ultra-high-performance concrete (UHPC). They found that, when the NS content was greater than 3% of the cement by weight, neither strength nor transport performance was improved, owing to the agglomeration of NS particles. Kooshafar and Madani [60] reported that the incorporation of NS into cement composites significantly reduced workability. This reduction was caused by strong agglomeration of NS particles within the cement composite. Wu et al. reported that a dry NS powder content of 1% in UHPC can improve the interfacial bonding between cement matrices, but a higher content may degrade performance instead [61]. As a result of the analysis of existing studies, there are still problems for application to construction products, such as dispersibility and optimal usage of nanomaterials.

The purpose of this study was to investigate the effects of NS and SF on the hydration reaction of HVFC. The main focus of this study is to solve the dispersibility problem caused by the agglomeration of NS powder in previous studies. To this end, NS was prepared in the form of a suspension so that the nano-silica particles were not physically agglomerated with each other. In addition, SF having a larger particle size than NS was mixed with NS to prepare an NSF suspension with a multiple particle size distribution. These two solutions were applied to an HVFC paste in which 50% of the cement volume was replaced with FA. The main variables were the NS and SF content. The hydration properties were evaluated through the setting time, heat of hydration, and compressive strength. In addition, the refinement of the pore structure by NS was analyzed using mercury intrusion porosimetry (MIP).

## 2. Materials and Methods

### 2.1. Materials

For this study, ordinary Portland cement (OPC), FA, SF, and NS were used as binders. The chemical compositions of the raw materials were analyzed using an X-ray fluorescence (XRF) spectrometer (ZSX Primus II Rigaku, Tokyo, Japan), with the results as shown in Table 1. The main components of the FA were SiO_2_ (50.5%), Al_2_O_3_ (24.3%), CaO (9.93%), and Fe_2_O_3_ (5.96%), corresponding to class F according to the ASTM C618 specifications. The main component of NS is SiO_2_ (99.9%). The densities of OPC, FA, SF, and NS are 3.16, 2.32, 2.26, and 2.23 g/cm^3^, respectively.

Figure 1 shows the X-ray diffraction (XRD) patterns for FA, SF, and NS. From the results in Figure 1, it was found that the main minerals constituting FA are quartz and mullite and that SF and NS are mostly amorphous. Both SF and NS have an amorphous structure that causes a centered halo corresponding to the main line of cristobalite.

Figure 2 shows the particle size distributions for OPC, FA, and SF. The average particle diameters are 19.1, 26.7, and 0.148 μm, respectively. As can be seen in the figure, the particle size distributions for OPC and SF are unimodal, whereas that for FA is bimodal. In general, the particle size distribution of fly ash varies depending on the type of fuel coal, combustion temperature, and process.

For the study, two types of solution were prepared and used to accelerate the hydration of an FA–cement mixture. The first was an NS solution with a solids content of 5%. The other, called the NSF solution, was prepared by mixing NS powder and SF at a weight ratio of 1:8; it had a solids content of 32%. Figure 3 shows SEM images of NS and NSF particles. The average diameter of NS particles measured from Figure 3a is 100 nm. In the case of NSF, as shown in Figure 3b, it was confirmed that the NS particles were evenly distributed around the SF particles.

### 2.2. Mix Proportions and Specimen Preparation

Cement pastes were prepared using the NS and NSF solutions, as shown in Table 2. The water-to-binder ratio was fixed at 0.3 for all specimens. For specimen Plain, 50% of the OPC volume was replaced by FA. For specimens NS05 and NS15, NS solution was added such that the NS solids content would be 0.5% and 1.5%, respectively, of the OPC volume. For specimen NS05SF4, NSF solution was added such that the NS and SF content would be 0.5% and 4%, respectively, of the OPC weight. The amounts of water added were determined based on the water content of the NS and NSF solutions.

After OPC and FA were placed in a forced mixer and mixed for approximately 30 s, NS or NSF solution was added, and the result was mixed at a speed of approximately 100 rpm for approximately 60 s, and then at approximately 200 rpm for 5 min. The paste specimen was then poured into a mold of size 40 mm × 40 mm × 160 mm and cured in a constant temperature and humidity chamber at a temperature of 20 ± 1 °C and a relative humidity of 90% or higher for one day. It was then demolded and subjected to water curing at 20 ± 1 °C until the test measurement date.

### 2.3. Test Methods

Figure 4 show the schematic diagram of the test process. The compressive strength of each paste specimen was measured at 3, 7, and 28 days of age in accordance with ISO 679. The specimen was split into two pieces for the flexural strength measurement test, and the compressive strength of each piece was measured. For each variable, the compressive strengths of the six samples were measured at each age, and the average values were taken as the results. The setting time was measured using an Acmel PA8 automatic setting time tester (ACMEL LABO, Saint Pierre du Perray, France) in accordance with ISO 9597. The paste for the setting time test was prepared in a constant temperature and humidity chamber at a temperature of 23 °C and a humidity of 60%. Initial setting and final setting were measured using an Acmel PA8 automatic setting time tester.

The hydration heat flow for each variable was measured using a TAM Air isothermal calorimeter (TA instruments, New castle, US). After a paste was prepared for each variable according to the mix proportions in Table 2, approximately 4 g of the paste was inserted into a glass ampule and then placed in the isothermal calorimeter. The temperature of the calorimeter was set to 23 °C, and measurements were performed every minute for 48 h.

MIP analysis was conducted using a Micromeritics AutoPore IV 9500 (Micromeritics, Norcross, US) to measure the pore size distribution and cumulative porosity of each specimen. At each measurement age, a sample was collected from the center of the specimen, and hydration was stopped. The sample was then vacuum-dried and stored to prevent contamination by carbonation. The pressure of mercury intrusion varied from 0 to 30,000 psi.

## 3. Results and Discussion

### 3.1. Hydration Properties

Figure 5 shows the penetration depth of the Vicat needle over time. Based on these results, the initial setting times for Plain, NS05, NS15, and NS05SF4 were measured as 5.92, 4.45, 3.97, and 3.82 h, respectively. Thus, the initial setting times for NS05, NS15, and NS05SF4 were reduced by 24.8%, 32.9%, and 35.5%, respectively, from that for Plain. The measured final setting times for Plain, NS05, NS15, and NS05SF4 were 6.89, 5.09, 4.93, and 4.74 h, respectively, similar to the results for the initial setting time. These results are in agreement with the finding by Chithra et al. [62] that NS promotes the setting of cement paste and decreases the dormant period of hydration by accelerating the hydration reaction of cement. Similar results were also confirmed in the study results of Bhatta et al. [63]. They reported that the initial setting time of cement composites decreased with increasing NS content. This result is because NS particles with large specific surface area provided more sites for hydration reaction. The setting time reduction was most obvious in NS05SF4, which contained both NS and SF. This appears to be because NS and SF further improved the filling rate by filling the gaps between cement particles and further accelerated the setting reaction by making the cement microstructure denser [64].

Figure 6 shows the hydration heat flow and cumulative heat release for each paste specimen. As can be seen from Figure 6a, the second heat flow peak occurred at 15.85, 11.03, 9.97, and 8.25 h for Plain, NS05, NS15, and NS05SF4, respectively. As the NS content increased, the peak occurred earlier and the magnitude of the peak tended to increase. Similar to the results for setting time (Figure 5), the second heat flow peak occurred earliest for NS05SF4, approximately 7.6 h earlier than that for Plain.

The cumulative heat release results (Figure 6b) show that the cumulative heat release for specimens containing NS was substantially greater than that for Plain until 32 h of hydration. This is in agreement with the finding by Xi et al. [65] that NS and SF particles accelerate hydration by acting as additional nucleation sites for cement hydrates during the initial stage of hydration. In particular, initial hydration was accelerated more strongly for NS05SF4 than for NS05 or NS15. For NS05SF4, however, the cumulative heat release became less than that for NS05 or NS15 after approximately 22 h of hydration. These results are consistent with those of Wang et al. [66]. They reported that when 0.5% of NS was added and 5% of SF was added, the cumulative heat release tended to decrease after 24 h.

As shown in Figure 7, NS with nano-sized particles fills the pores between the cement particles during the initial hydration process, thereby causing the structural filler effect inside the cement. This can improve the density of the matrix by effectively refining capillary pores and decreasing the porosity of the matrix [65]. Pacheco-Torgal et al. [67] reported that the addition of NS makes the internal structure of concrete mortar denser.

In addition, NS particles accelerate the cement hydration reaction by providing more nucleation sites for the precipitation of hydration products during the initial hydration process (Figure 7). NS particles preferentially adsorb the C-S-H gels generated during hydration through their large specific surface area. The C-S-H gels adsorbed on the surface of NS particles act as crystal nuclei as they propagate between the cement particles [65]. This increases strength and promotes setting by further accelerating the cement hydration reaction and forming more C-S-H gels. Silvestre et al. found that NS particles that act as nucleation sites form a larger network through connection with the C-S-H network in the cement system and improve mechanical performance [23].

### 3.2. Compressive Strength

Figure 8 shows the compressive strengths of the specimens according to age. The compressive strengths of all non-Plain specimens were found to be higher than that of Plain at all ages, except for the compressive strength of NS05 at 7 days. At 3 days, the compressive strengths of NS05, NS15, and NS05SF4 were 105%, 103%, and 116% that of Plain. This is due to the filler effect, in which NS fills empty spaces between the cement particles during the initial stage of hydration, accelerating the hydration reaction by providing nucleation sites [51]. These results are similar to those of Chekravarty et al. [68]. They reported that the compressive strength increased up to 3% of the NS content, which was attributed to the improvement of the microstructure and aggregate–paste interface by NS. At 28 days, the compressive strengths of NS05, NS15, and NS05SF4 were 106%, 110%, and 110% that of Plain. This is because of the acceleration of the hydration reaction by the additional pozzolanic reactions of NS and SF particles during the middle stage of hydration.

NS and SF particles can generate C-S-H gels through fast reactions with calcium hydroxide (Ca(OH)_2_), known as the pozzolanic reaction [65]. NS and SF particles that have interpenetrated the cement particles elute SiO_4_^4−^ and AlO_2_^−^ ions as the hydration reaction continues. They react with the Ca^2+^ ions eluted from the C_3_S particles to form C-S-H gels around the pozzolan particles. This pozzolanic reaction of NS and SF improves long-term strength during the cement hydration process [69]. Hanif et al. reported that the pozzolanic activity can be significantly accelerated by adding NS particles to FA concrete composites [70].

Figure 9 shows the compressive strength increase for the specimens according to age. In general, the strength improvement effect of the C-S-H gels generated by the pozzolanic reaction occurs after 3 or 7 days of age. Thus, the effect of the additional pozzolanic reactions of NS and SF particles during the middle stage of hydration could be assessed by measuring the change in compressive strength after 3 or 7 days. As can be seen from Figure 9, during the interval from 3 to 7 days, the compressive strengths of Plain, NS05, NS15, and NS05SF4 increased by 11.17, 8.58, 12.74, and 10.64 MPa, respectively. Except for NS15, the strength increases of the non-Plain specimens were slightly lower than that of Plain. During the interval from 3 to 28 days, the compressive strengths of Plain, NS05, NS15, and NS05SF4 increased by 25.69, 27.54, 31.15, and 26.57 MPa, respectively, and from 7 to 28 days, the compressive strengths increased by 14.52, 18.96, 18.41, and 15.93 MPa, respectively. After 7 days of age, the strength increase of all non-Plain specimens was greater than that of Plain. The greatest increase was observed during the interval from 7 to 28 days, and the slope of the strength increases for all non-Plain specimens was higher than that for Plain. This is because the pozzolanic reaction was accelerated by NS and SF particles that interpenetrated the cement particles during the hydration process.

### 3.3. Microstructure

Figure 10 shows the log differential intrusion for each specimen by pore diameter according to age. Three experiments were conducted for each variable, and the average value was used. The standard deviation was less than 6%.

For all specimens, pore size tended to decrease as age increased. At 1 day, the main pore size of Plain ranged from 500 to 1000 nm. For the other specimens, however, the peaks in pore size occur in approximately the 300–700 nm range. At 3 and 7 days, the pore sizes of all specimens tended to be less than that at 1 day. This appears to be due to the filler effect of NS and SF particles. At 28 days, pore diameters of the non-Plain specimens corresponding to the peaks were substantially less than those of Plain. In particular, the peak in pore size occurs in approximately the 150–200 nm range for NS15 and NS05SF4; this is substantially smaller than for Plain, whose pore sizes peak in the 300–400 nm range. This appears to be due to the improvement in pore structure caused by the pozzolanic activity of NS and SF particles during the middle stage of hydration. Miao et al. [71] reported that NS particles optimize the pore structure and replace large pores with small pores. They reported that NS solution containing PCE (polycarboxylate ether) can effectively improve the strength of cement mortar even with a very small amount.

Figure 11 shows the cumulative pore volume of each specimen according to age. As age increased, the pore volumes of all specimens decreased. At 1 day, the cumulative pore volumes of Plain, NS05, NS15, and NS05SF4 were 0.157, 0.165, 0.161, and 0.167 cm^3^/g, respectively. Although Plain showed the lowest value, the difference was negligible. At 3 days, however, a different pattern was observed. By 7 days, the difference was pronounced: the cumulative pore volumes of Plain, NS05, NS15, and NS05SF4 were 0.124, 0.121, 0.096, and 0.114 cm^3^/g, respectively. Plain showed the highest value, and the value of NS15 was 77.8% that of Plain. These results display a pattern similar to that in the compressive strength increase results for the interval from 3 to 7 days of age. Then, at 28 days, the pore volumes of the non-Plain specimens were less than those of Plain, with the pore volume of NS05SF4 having the lowest value (0.084 cm^3^/g). This again is similar to the compressive strength results and appears to be due to the formation of additional reaction products caused by the additional pozzolanic reaction activation of NS and SF particles that penetrated into the cement composite after 7 days. The ratios of the pore volumes at 28 days compared with 7 days for Plain, NS05, NS15, and NS05SF4 were 81.2%, 81.1%, 96.0%, and 73.3%, respectively. NS15 displayed the lowest reduction in pore volume, and NS05SF4 exhibited the highest. This appears to be due to the effect of the pozzolanic reaction for SF with larger particle sizes.

## 4. Conclusions

This study investigated the effects of nano-silica (NS) and silica fume (SF) on the hydration reaction of high-volume fly ash cement (HVFC). The main findings of the study are as follows.

The initial and final setting times of HVFC were shortened as the NS and SF particle content increased. The setting time was shorter when NS and SF were added together than when NS was used alone. In particular, in the case of the specimen containing the NSF solution with multiple particle sizes, the initial setting time was reduced by 35.5% from that of the plain specimen. This appears to be the result of the NSF solution with multiple size distribution filling the gaps between the cement particles and improving the filling rate.The peak occurrence time of hydration heat flow decreased and the magnitude of the peak increased as the NS content increased. The cumulative heat release of all non-plain specimens was higher than that of the plain specimen until 22 h of hydration. This appears to be because NS and SF particles acted as additional nucleation sites for cement hydrates during the initial stage of hydration. In the case of the specimen containing the NSF solution, however, the cumulative heat release became less than that for NS05 or NS15 after approximately 22 h of hydration. These results are similar to the previous studies that reported that the cumulative caloric value decreased after 1 day of age when NS and SF were added together. Therefore, future studies on optimal NS and SF content are needed.When NS and SF were mixed, the compressive strength of the cement composite showed a tendency to improve. The compressive strengths of all specimens were higher than that of the Plain at all ages, except for the compressive strength of NS05 at 7 days of age. In addition, the increase in the compressive strength of all non-plain specimens after 7 days was greater than that of the Plain, the largest increase being in the age interval from 7 to 28 days. This appears to be because the pozzolanic reaction was accelerated by NS and SF particles. A quantitative study on the effect of pozzolanic reaction of NS and SF on long-term compressive strength is needed in the future.The pore size of the specimens decreased according to NS and SF particle content. After 28 days of age, the pore diameters of the non-plain specimens were substantially less than those of the Plain. This appears to be due to the improvement in pore structure caused by the pozzolanic activity of NS and SF particles during the middle stage of hydration. In addition, the cumulative pore volumes of all specimens decreased after 7 days of age, with the greatest decrease being that at 28 days in the specimen containing the NSF solution.NS and SF improved the early strength of cement composites and improved the internal pore structure. In particular, the mechanical properties and pore structure of the cement composite were further improved when NS and SF were mixed together than when NS was used alone. It seems that the properties of the cement composite will vary depending on the amount of NS and SF mixed. Therefore, in order to apply nanomaterials such as NS and SF to construction products, research on the optimum mix design suitable for the characteristics of each construction product is required in the future.

## Figures and Tables

**Figure 1 materials-15-06599-f001:**
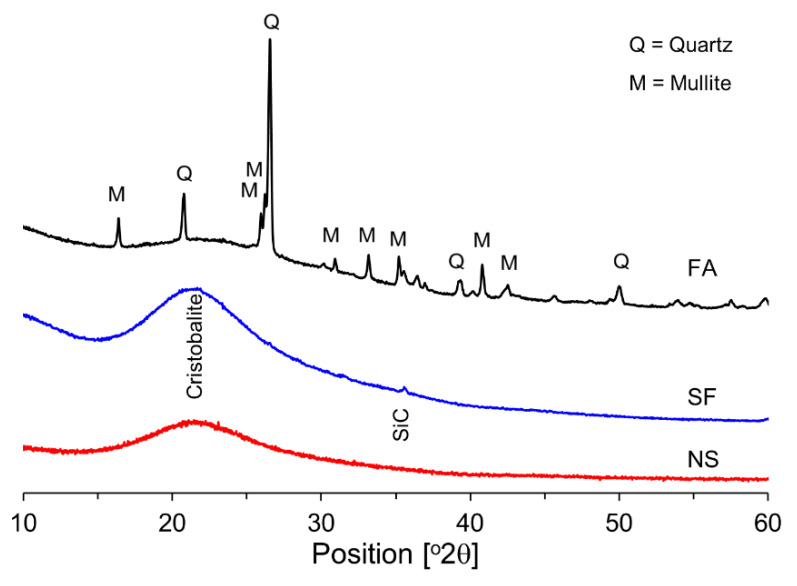
XRD patterns of FA, SF, and NS.

**Figure 2 materials-15-06599-f002:**
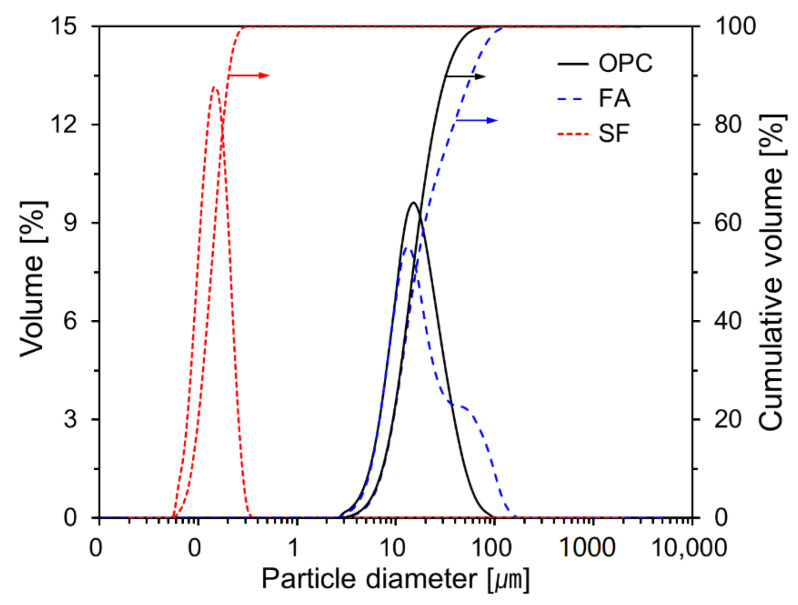
Particle size distributions for OPC, FA, and SF.

**Figure 3 materials-15-06599-f003:**
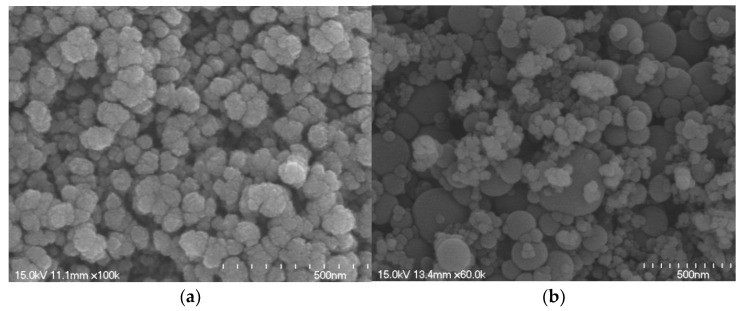
SEM images of (**a**) NS solution (×100,000) and (**b**) mixed NS/SF (NSF) solution (×60,000).

**Figure 4 materials-15-06599-f004:**
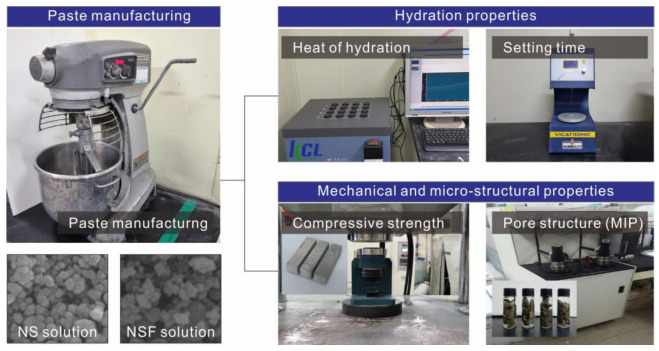
Schematic diagram of the test process.

**Figure 5 materials-15-06599-f005:**
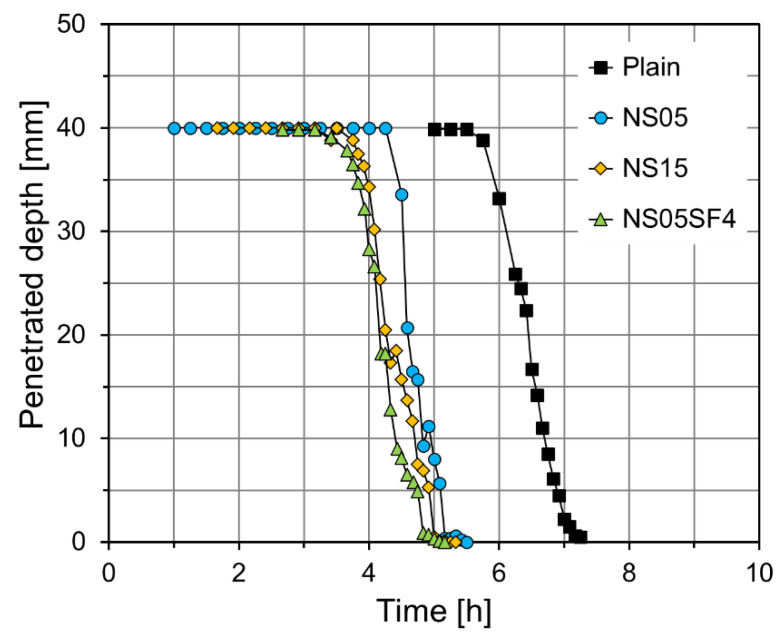
Results of Vicat needle test on the specimens.

**Figure 6 materials-15-06599-f006:**
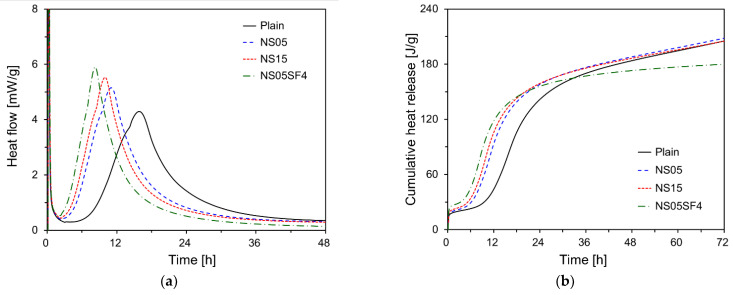
(**a**) Hydration heat flow and (**b**) cumulative heat release.

**Figure 7 materials-15-06599-f007:**
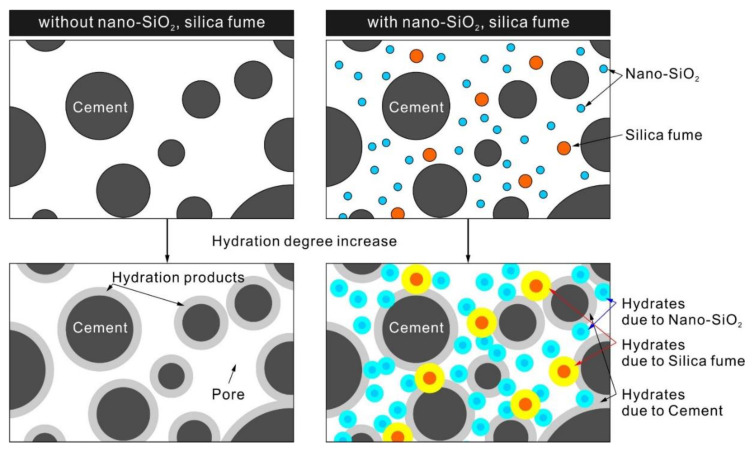
Hydration reaction caused by the NS and SF particles.

**Figure 8 materials-15-06599-f008:**
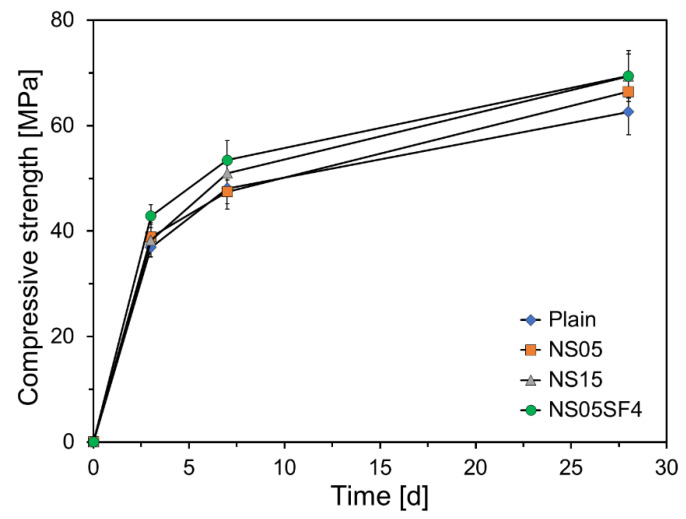
Compressive strengths of specimens.

**Figure 9 materials-15-06599-f009:**
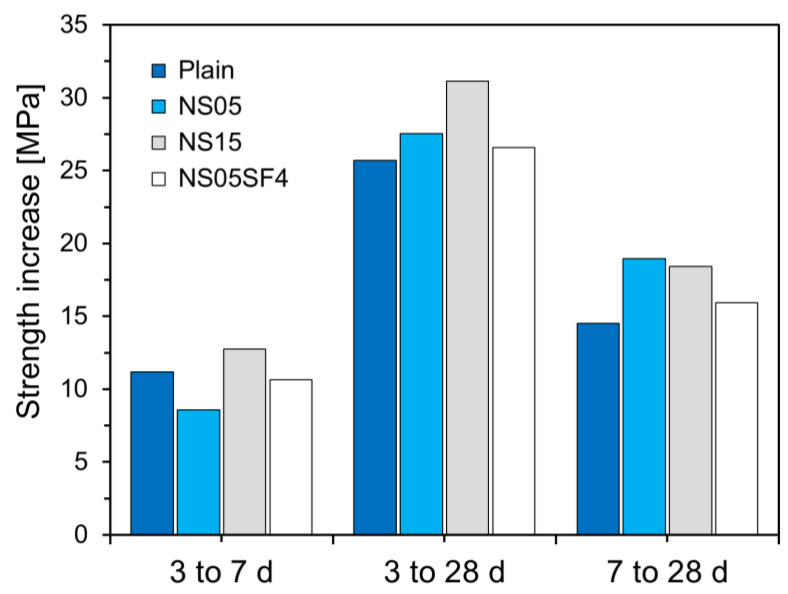
Increases in specimen strengths during different age intervals.

**Figure 10 materials-15-06599-f010:**
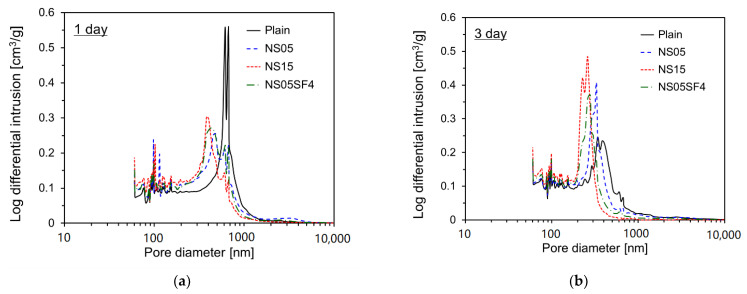
Log differential intrusions for specimens by pore diameter. (**a**) 1 day; (**b**) 3 days; (**c**) 7 days; (**d**) 28 days.

**Figure 11 materials-15-06599-f011:**
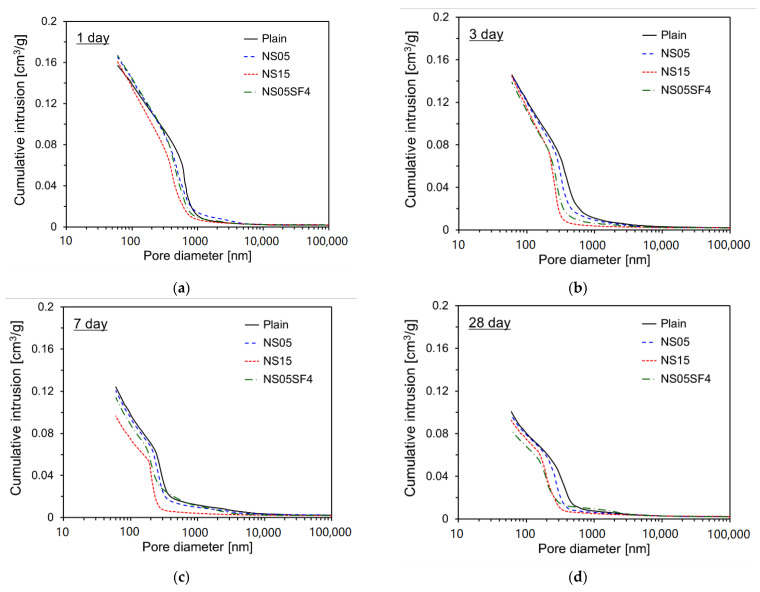
Cumulative intrusions in specimens. (**a**) 1 day; (**b**) 3 days; (**c**) 7 days; (**d**) 28 days.

**Table 1 materials-15-06599-t001:** Chemical composition of binders.

Binder	Chemical Composition (wt.%)									
	CaO	SiO_2_	Al_2_O_3_	Fe_2_O_3_	MgO	TiO_2_	K_2_O	Na_2_O	SO_3_	LOI
OPC	63.2	20.2	4.1	3.6	2.7	0.2	1.0	0.1	1.4	0.8
FA	9.93	50.5	24.3	6.0	1.5	1.2	1.6	0.9	2.2	1.8
SF	0.2	91.6	0.4	0.9	1.2	–	1.0	0.8	0.4	0.6

Abbreviations: FA, fly ash; OPC, ordinary Portland cement; SF, silica fume.

**Table 2 materials-15-06599-t002:** Mix proportions of pastes.

Specimen	W/B	Binder (g)				SP (wt.% by Binder)
		OPC	FA	NS	SF	
Plain	0.3	100	73.42	0	0	0.06
NS05		99	73.42	0.71	0	0.25
NS15		97	73.42	2.12	0	0.62
NS05SF4		91	73.42	0.71	5.72	0.30

Abbreviations: W/B, water-to-binder ratio

## Data Availability

The raw/processed data will be provided as request.

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
