# Peer review of "Hydration and Mechanical Properties of High-Volume Fly Ash Concrete with Nano-Silica and Silica Fume"

_materials, 2022, doi:10.3390/ma15196599_

Round 1

Reviewer 1 Report

This Manuscript investigate the  “Hydration and mechanical properties of high-volume fly ash concrete with nano-silica and silica fume”. The introduction section provides sufficient background of past literatures. In the experimental Programme section, all the testing methods are sufficiently described. In the experimental result and discussion section, the results are clearly depicted with figures and tables. The conclusions are well presented. All the references are related to this research. However, the following minor corrections are to be carried before the acceptance of the Manuscript.

1. Arrange the Keywords in alphabetical order

2.Introduction: The sentence “Accordingly, studies have been conducted on a variety of construction materials, and the use of supplementary cementing materials (SCMs), such as fly ash (FA), ground granulated blast-furnace  slag (GGBS), and silica fume (SF), has attracted attention” need some citations. One such a work is shown below.

https://doi.org/10.1016/B978-0-12-821730-6.00031-0

3.Introduction.

Yang et al. reported that adding SF to Na2SO4-activated high-volume fly ash (HVFA) concrete leads to higher resistance and less strength loss [14].

The citation [14] should be next to Author, but, not at the end. Also, Cite in similar manner throughout the manuscript.

4. State the novelty of your research at the end of the introduction.

5. Correct table 1 caption as “Chemical composition of binders”

6. Correct table 2 caption as  “Mix proportions of pastes”

7. The conclusion part simply reported the findings. The conclusion should be in such a way that, it should be supported by  the result obtained from the study. Also, give your recommendation at the end of the conclusion

Reviewer 2 Report

This paper presents about the hydration and mechanical properties of high-volume fly ash concrete with NS and AF, which is of great significance to HVFC. However, the contents of this paper were focused on "test" in which it looks like an experimental report. In addition, there are still some questions in the article that need the author to reply and answer.

 1. In Section 1, only few studies are mentioned in the introduction (line 41-68) without any discussion of the advantages and the drawbacks of each study. I recommend to add recent references with investigation. In addition, the article needs to analyze the shortcomings of the existing research.

 2. In Section 1, the last paragraph of the introduction does not indicate the key problems and innovations to be solved in the study. It needs to be supplemented according to the shortcomings of existing studies.

 3. In line 95-96 of Section 2.1, it is proposed that FA is a bimodal distribution, and the reason should be explained.

 4. Fig. 3 shows the SEM image, but no analysis is performed. What is the meaning of Figure 3? It is recommended to delete or analyze the graph.

 5. In Section 2.2, what is the basis for the paste composition ratio to be tested? Why is this configuration required?

 6. In Section 2.3, it is suggested to add pictures of the test process, including the physical picture of the sample and the instrument.

 7. The conclusion parts should be carefully rewritten to elaborate the main contributions.

 8. Include future work scope at the end of the conclusion section.

 9. This paper has carried out a lot of experimental analysis on HVFC, but it does not reflect the innovation and scientific problems. The study need proper scientific discussions.

Reviewer 3 Report

Please further elaborate on the novelty of your work in the abstract.

The presented introduction is pretty modest. Please include a brief but critical review regarding the conducted research studies in the introduction.  It is recommended to add a section “research significance” and highlight the main contribution of your findings.

Please provide a brief summary of using nano silica particles (NS) on the mechanical properties of cement paste in concrete using a brief summary of the article titled Mechanical Characteristics of Cement Paste in the Presence of Carbon Nanotubes and Silica Oxide Nanoparticles: An Experimental Study.

Please include the latest research studies related to your work preferably between 2019 and 2022

Please review the latest studies on using fly ash in concrete and its effect on mechanical properties of concrete using the work titled Predicting the Effect of Fly Ash on Concrete’s Mechanical Properties by ANN.

Please further explain the SEM images in Figure 3 of to illustrate the characteristics of concrete by adding nano silica and slica fume to show the distribution and hydration characteristics of the tested specimen.

Please identify the main parameters that can affect the result of your work.

Please include statistical characteristics of the reported curves based on the presented data in Figures 11 and 12.

Please discuss the synergistic effect of using nano silica and silica fume with different percentages.

Please include a quantitative approach for reporting the test outcomes in the conclusion.

Please highlight the shortcomings in this research study and include recommendations for future research.

Round 2

Reviewer 2 Report

The authors answered all the questions and revised the paper. But I think the revised paper still needs moderate English changes.

Reviewer 3 Report

N/A